# Efficient Cell Painting Image Representation Learning via Cross-Well Aligned Masked Siamese Network

## Abstract

Computational models that predict cellular phenotypic responses to chemical and genetic perturbations can accelerate drug discovery by prioritizing therapeutic hypotheses and reducing costly wet-lab iteration. However, extracting biologically meaningful and batch-robust cell painting representations remains challenging. Conventional self-supervised and contrastive learning approaches often require a large-scale model and/or a huge amount of carefully curated data, still struggling with batch effects. We present Cross-Well Aligned Masked Siamese Network (CWA-MSN), a novel representation learning framework that aligns embeddings of cells subjected to the same perturbation across different wells, enforcing semantic consistency despite batch effects. Integrated into a masked siamese architecture, this alignment yields features that capture fine-grained morphology while remaining data- and parameter-efficient. For instance, in a gene-gene relationship retrieval benchmark, CWA-MSN outperforms the state-of-the-art publicly available self-supervised (OpenPhenom) and contrastive learning (CellCLIP) methods, improving the benchmark scores by +29% and +9%, respectively, while training on substantially fewer data (e.g., 0.2M images for CWA-MSN vs. 2.2M images for OpenPhenom) or smaller model size (e.g., 22M parameters for CWA-MSN vs. 1.48B parameters for CellCLIP). Extensive experiments demonstrate that CWA-MSN is a simple and effective way to learn cell image representation, enabling efficient phenotype modeling even under limited data and parameter budgets. The source code for CWA-MSN is available at anonymous code link.

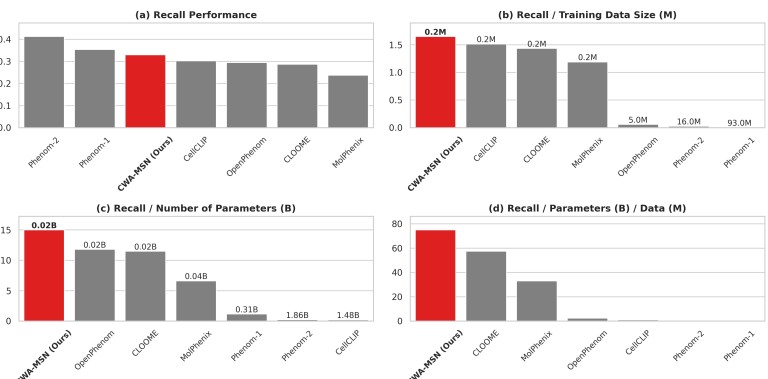

Figure 1: **Comparison of methods based on gene-gene interaction benchmark over multiple efficiency metrics:** (a) benchmark results measured as recall, (b) recall normalized by training data size (per million images), (c) recall normalized by number of parameters (per billion), and (d) recall normalized by the product of both training data size and number of parameters. Our method for each metric plot is highlighted in red. Annotated values indicate either training dataset size (M) or number of parameters (B). Except (a), CWA-MSN is top-performing, showcasing its data- and parameter-efficient learning for cell representation.

## 1    INTRODUCTION

Computational modeling of how cells respond to chemical and genetic perturbations is a promising strategy in drug discovery (Noutahi et al., 2025; Liu et al., 2025; Navidi et al., 2025). By predicting therapeutic effects and revealing potential mechanisms of action (Tanaka et al., 2025), these cell models help reduce the need for costly and time-consuming wet lab experiments (Bunne et al., 2024; Adduri et al., 2025). Furthermore, such predictive models are valuable tools for researchers, helping them generate targeted hypotheses and accelerating the transition from early-stage drug screening to experimental validation (Stokes et al., 2020). In particular, recent advances in high-content screening (HCS) (Bickle, 2010) enable automated, high-throughput acquisition of cell painting images across diverse perturbations (Starkuviene & Pepperkok, 2007), creating high-dimensional datasets that support phenotype-driven modeling of perturbation effects (Nierode et al., 2016).

However, significant challenges remain in extracting biologically meaningful representations from cell painting images. For instance, CellProfiler (Stirling et al., 2021), a widely used analysis tool, relies on predefined image features such as shape, intensity, and texture. Although it has facilitated numerous biological discoveries (Boutros et al., 2015; Ariffin, 2023), the method is highly susceptible to batch effects arising from various experimental conditions, including lighting, stain intensity, and instrument settings (Arevalo et al., 2024b). Moreover, because its features are handcrafted, CellProfiler cannot adapt or improve its phenotypic representations as more diverse data become available, limiting its ability to capture complex or subtle phenotypic variations (Kim et al., 2025).

Recent advances in self-supervised learning (SSL) (He et al., 2020; Chen et al., 2020; Chen & He, 2021; Caron et al., 2021; He et al., 2022) have been successfully applied to extracting characteristics of cell painting images, showing its potential to derive rich morphological information (Kraus et al., 2024; Kenyon-Dean et al., 2024). Nevertheless, these data-driven approaches depend on computationally intensive foundation models (Dosovitskiy et al., 2020) and, due to the absence of explicit labels, require extremely large and carefully curated phenotype-diverse data. As an alternative, weakly supervised and contrastive learning has been adopted, leveraging proxy labels (e.g., cell and perturbation types) as training signals for data-efficient learning (Moshkov et al., 2024; Caicedo et al., 2018; Lu et al., 2025; Sanchez-Fernandez et al., 2023; Fradkin et al., 2024; Bushiri Pwesombo et al., 2025). However, even with these advances, both approaches remain vulnerable to batch effects, which are derived from different experimental conditions.

In this work, we introduce the Cross-Well Aligned Masked Siamese Network (CWA-MSN), a novel representation learning framework for cell painting images. Unlike self-supervised methods, CWA-MSN leverages weak perturbation labels to align representations across wells of the same perturbation, which often exhibit varying batch effects. This alignment strategy enforces robust semantic consistency in the learned feature space, ensuring that biologically meaningful relationships are preserved despite experimental variability. By incorporating this cross-well alignment into a masked siamese network (Assran et al., 2022), CWA-MSN achieves substantial improvements in both capturing intricate phenotypic relationships and maintaining high data and parameter efficiency.

Extensive experiments demonstrate that CWA-MSN consistently outperforms existing approaches in biological relationship retrieval tasks, particularly in gene–gene and compound–gene associations (Kraus et al., 2025). For instance, CWA-MSN surpasses state-of-the-art (SOTA) publicly available self-supervised method, OpenPhenom (Kraus et al., 2024), and weakly-supervised contrastive method, CellCLIP (Lu et al., 2025), with gains of 29% and 9% in the gene–gene interaction benchmark. Moreover, CWA-MSN achieves these improvements with significantly reduced amount of training data (e.g., 0.2M images for CWA-MSN vs. 2.2M images for OpenPhenom) or much smaller model size (e.g., 22M parameters for CWA-MSN vs. 1.48B parameters for CellCLIP). Fig. 1 compares CWA-MSN with existing methods based on model size, training data, and benchmark performance, showcasing its strong advantages over the others.

## 2    RELATED WORK

### 2.1    SELF-SUPERVISED LEARNING FOR CELL PAINTING IMAGES

Recent success in self-supervised representation learning (He et al., 2020; 2022; Chen et al., 2020; Chen & He, 2021; Caron et al., 2021) has spurred interest in applying these methods to microscopy

images. However, some approaches face limitations when transferred from natural images to HCS images. For example, training process of DINO (Caron et al., 2021) relies on data augmentation strategies designed for natural images, which reduces its effectiveness on HCS data (Doron et al., 2023; Kim et al., 2025; Kraus et al., 2024). Masked image modeling methods, such as MAE (He et al., 2022), offer a better alternative by reducing dependency on data augmentation selection. Indeed, recent applications of MAE in HCS imaging (Kraus et al., 2024; Kenyon-Dean et al., 2024) have demonstrated impressive performance in retrieving known biological relationships between perturbations. However, these approaches require substantial computational resources (e.g., 256 H100 GPUs in Kenyon-Dean et al. (2024)) and large-scale curated training data (e.g., 93 million cell images in Kenyon-Dean et al. (2024)). To circumvent this problem, in this study, we propose a more data- and parameter-efficient approach which can still achieve competitive performance.

## 2.2 WEAKLY SUPERVISED AND CONTRASTIVE LEARNING FOR CELL PAINTING IMAGES

Weakly supervised learning (WSL) and contrastive learning (Yu et al., 2025; Bao et al., 2023), which are training methods utilizing proxy labels as guiding signals, have been adopted for the development of an image encoder for cell painting images (Moshkov et al., 2024; Caicedo et al., 2018; Sanchez-Fernandez et al., 2023; Lu et al., 2025; Fradkin et al., 2024; Bushiri Pwesombo et al., 2025). For example, SemiSupCon (Bushiri Pwesombo et al., 2025) jointly align the features of replicative treatment pairs using contrastive learning but not explicitly collating training data in different wells, plates, and batches. Also, CellCLIP (Lu et al., 2025) uses text encoding, such as perturbation and cell types, as proxy signals for training by aligning the image features together. Although SSLProfiler (Dai et al., 2025) also explores cross-image alignment by matching multiple site (e.g., within-well) images using DINOv2 with auxiliary branches and multiple losses, it assumes identical perturbations across corresponding well positions and only addresses site-level variation. This can conflict with known positional batch effects (Moshkov et al., 2024). In contrast, our method explicitly targets cross-well and cross-plate variation, aligning wells with the same perturbation in prototype space using a single, unified objective. Although the methods in this category are mostly data-efficient and show promising results, they often conflate confounding factors (e.g., batch effects) with true phenotypic outcome because it is generally impossible to explicitly derive all the components that influence the true perturbation effects (e.g., all causes of batch effects) as weak labels. In this work, we leverage a cross-well alignment strategy which can naturally overcome this limitation without relying on explicit definition of proxy labels as additional training signals.

## 3 CROSS-WELL ALIGNED MASKED SIAMESE NETWORK

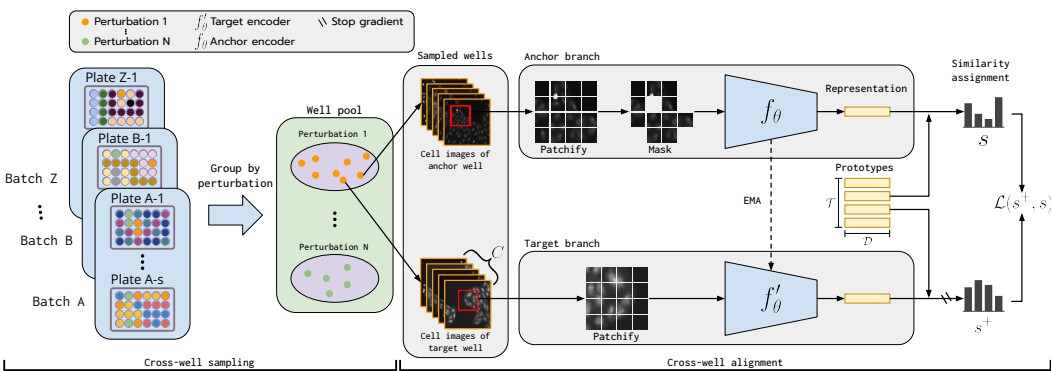

Figure 2: **Overview of CWA-MSN framework:** The framework is composed of two parts, cross-well sampling and cross-well alignment. Cross-well sampling selects cell images under the same perturbation from different wells across batches and plates to serve as an implicit data augmentation strategy. Cross-well alignment uses a masked siamese network to align anchor and target well representations by matching their prototype-based similarity distributions.

## 3.1 PROBLEM STATEMENT

HCS experiments generate hierarchically organized cellular imaging data. A batch typically corresponds to a collection of plates (e.g., see Batch A in Fig. 2) processed under uniformly controlled experimental conditions (e.g., each batch for each day), and a plate consists of multiple wells (e.g., 96, 384, etc.) containing replicative measurements of cells subjected to a specific perturbation (e.g., six replicative wells per perturbation).

During HCS experiments, batch effects are introduced by several factors, such as systematic differences in instrumentation settings, imaging time and conditions, sample preparation, and technical noise. These unintended factors can obscure the true biological signal associated with each perturbation by disturbing the phenotypic representations of cells and, consequently, make it difficult to uncover actual biologically relevant changes. To address this challenge, we propose a data-efficient approach that captures true phenotypic perturbation differences and mitigates batch effects by using cross-well alignment and masked siamese network learning.

## 3.2 CROSS-WELL SAMPLING

In the cross-well sampling of CWA-MSN, we aim to utilize cross-well images of the same perturbation across different batches and plates as an implicit data augmentation strategy. We detail the sampling strategy in the following.

Let $P = \{p_1, p_2, \ldots, p_N\}$ denote the set of $N$ chemical or genetic perturbations (e.g., compounds, gene knockouts). Each perturbation $p_i$ is associated with a set of wells in different plates and batches (see batch A to Z in Fig. 2). Also, each well can be scanned multiple times using different staining methods (e.g., Hoechst, Phalloidin).

We define a set of wells for the perturbation $p_i$ as follows:

$$W_i = \{w_1^{(i)}, w_2^{(i)}, \ldots, w_{M_i}^{(i)}\}, \quad w \in \mathbb{R}^{C \times H \times W},$$

where $M_i$ is the total number of wells under perturbation $p_i$, $w$ refers to cell images of a single well, $C$ is the number of staining channels, and $H \times W$ is the spatial dimension of a single cell image. For cross-well sampling, we first randomly choose a single perturbation $p \in P$, then select cell images of two distinct wells under the same perturbation:

$$w_a^p, w_t^p \in W_p, \quad w_a^p \neq w_t^p,$$

Here, $w_a^p$ and $w_t^p$ are designated as cell images of *anchor well* and *target well*, respectively (see Sampled Wells in Fig. 2). During the sampling procedure, it is possible that these two wells are from the same plate or two separate plates in different batches.

## 3.3 CROSS-WELL ALIGNMENT VIA MASKED SIAMESE NETWORK

Next, we combine the above cross-well sampling strategy with a masked siamese network Assran et al. (2022). In contrast to MAE, which masks part of *a single image* and trains a model to recover it, MSN aligns the representations of *two images* (cross-well images in our case) with their masked and unmasked counterparts through prototype-based learning (see Section 5.4 for the comparison results of CWA-MAE vs. CWA-MSN).

For the anchor well $w_a^p$, we construct multiple augmented views applying random cropping and flipping, following the procedure of Assran et al. (2022):

$$\mathbf{X}_a^{(p)} \in \mathbb{R}^{V_a \times C \times H \times W},$$

where $V_a$ denotes the number of augmentations. For the target well $w_t^p$, we generate a single augmented view:

$$\mathbf{X}_t^{(p)} \in \mathbb{R}^{1 \times C \times H \times W}.$$

From now on, $\mathbf{X}_a^{(p)}$ and $\mathbf{X}_t^{(p)}$ are designated as *anchor view* and *target view*, respectively. Then, a mini-batch for training is collated through stacking the anchor and target views of a set of perturbations $P_B \subset P$ where $|P_B| = B$, and $B$ is the number of perturbations within the mini-batch:

$$\mathbf{X}_a = \{\mathbf{X}_a^{(p)}\}_{p \in P_B} \in \mathbb{R}^{B \times V_a \times C \times H \times W}, \quad \mathbf{X}_t = \{\mathbf{X}_t^{(p)}\}_{p \in P_B} \in \mathbb{R}^{B \times 1 \times C \times H \times W}.$$

We process the anchor view $\mathbf{X}_a$ by patchifying and masking with ratio $\alpha$, whereas the target view $\mathbf{X}_t$ is simply patchified. After that, $z$ and $z^+$, representing the embeddings of the anchor and target views, are extracted through the anchor encoder $f_\theta$ and the target encoder $f'_\theta$, respectively (see Fig. 2).

$$z = f_\theta(\mathbf{X}_a) \in \mathbb{R}^{B \times V_a \times \mathcal{D}}, \quad z^+ = f'_\theta(\mathbf{X}_t) \in \mathbb{R}^{B \times 1 \times \mathcal{D}}, \tag{1}$$

where $\mathcal{D}$ denotes the representation dimension of each view and $B$ is the batch size. With a set of prototype embeddings $\mathrm{O} \in \mathbb{R}^{\mathcal{T} \times \mathcal{D}}$, where $\mathcal{T}$ is the number of prototypes, we compute the similarity assignment scores as

$$s = \mathrm{sim}(\mathrm{O}, z) \in \mathbb{R}^{B \times V_a \times \mathcal{T}}, \quad s^+ = \mathrm{sim}(\mathrm{O}, z^+) \in \mathbb{R}^{B \times 1 \times \mathcal{T}}, \tag{2}$$

where $\mathrm{sim}(\cdot, \cdot)$ denotes the normalized cosine similarity as in Assran et al. (2022). Finally, the model is trained by aligning the similarity distributions of the anchor and target views, $(s, s^+)$ with an auxiliary mean entropy maximization to prevent the collapse of the features as follows:

$$\mathcal{L}(s^+, s) = \lambda_1 \, CE(s^+, s) + \lambda_2 \frac{1}{B\mathcal{T}} \sum_{j=1}^{B} \sum_{m=1}^{\mathcal{T}} s_{j,m}, \tag{3}$$

where $CE(\cdot, \cdot)$ is a cross-entropy loss and $\lambda_1, \lambda_2$ are balancing coefficients. For simplicity, we omit the summation $s$ over $V_a$ in this definition. During training, the target encoder, $f'_\theta$, is not directly updated through backpropagation, but indirectly updated by an exponential moving average (EMA) of model weight of the anchor encoder, $f_\theta$, (see Stop gradient and EMA in Fig. 2).

### 3.4 IMPLEMENTATION DETAILS

We use ViT-S/16 as both anchor and target encoders and train with batch size 64 for 100 epochs using AdamW. The initial learning rate is 0.0002, following a cosine decay schedule with a 15-epoch warm-up. Weight decay increases from 0.04 to 0.4 via cosine schedule. We set $\mathcal{T} = 1024$ prototypes, representation dimension $\mathcal{D} = 256$, and loss weights $\lambda_1 = \lambda_2 = 1$. The anchor masking ratio is $\alpha = 0.15$. Ablation study of $\alpha$ is further provided in Appendix A.1. We follow Assran et al. (2022) for anchor views, using one random and ten focal crops ($V_a = 11$). The target encoder is updated via EMA with momentum starting at 0.996 and linearly increasing to 1.0.

## 4 EXPERIMENTS

### 4.1 TRAINING DATA OF CWA-MSN

For the development of CWA-MSN, we utilized a Bray dataset (Bray et al., 2016) which encompasses five-channel cell painting images perturbed by diverse small-molecules. First, we applied a preprocessing pipeline to cell images as described in Sanchez-Fernandez et al. (2023). Then, following the setting in CellCLIP (Lu et al., 2025), we selected 70% of the total data for training, including 198,609 cell images with 7,401 distinct perturbations. Note that the size of the training data (that is, 0.2 M) is much smaller than that of recent self-supervised methods (from 5M to 93M; see Fig. 1).

### 4.2 BENCHMARKS

**Gene-Gene Interaction Benchmark** RxRx3-core (Kraus et al., 2025) is a curated benchmark dataset to evaluate zero-shot performance of a cell painting image encoder, circumventing the limitations of existing benchmarks (i.e., CPJUMP1 (Chandrasekaran et al., 2024) and Motive (Arevalo et al., 2024a)) such as small perturbation coverage and biased well positions. The dataset consists of 1,335,606 images perturbed by 736 gene knockouts and 1,674 small-molecules.

In a gene-gene interaction benchmark of RxRx3-core (Celik et al., 2024), a model is evaluated by calculating pairwise cosine similarities between all feature of gene-gene pairs (e.g., MTOR and TSC2 genes) and selecting the pairs of the highest or lowest 5% similarity scores among them. After that, the selected positive (i.e., highest) or negative (i.e., lowest) relationships are compared with known biological gene-gene association databases, including Reactome, HuMAP, SIGNOR,

StringDB, and CORUM (Giurgiu et al., 2019; Drew et al., 2017; Gillespie et al., 2022; Szklarczyk et al., 2021). Recall values (i.e., discovered relationships / known relationships) were measured and reported for each database.

In Section 5.1, we have compared the performance of the proposed CWA-MSN with previous handcrafted (CellProfiler (Stirling et al., 2021)), weakly-supervised (SupCon (Khosla et al., 2020), MolPhenix (Fradkin et al., 2024), CLOOME (Sanchez-Fernandez et al., 2023), and CellCLIP (Lu et al., 2025)), contrastive learning (SimCLR (Chen et al., 2020)), and self-supervised methods (OpenPhenom (Kraus et al., 2024), Phenom-1 (Kraus et al., 2024), and Phenom-2 (Kenyon-Dean et al., 2024)). For additional comparisons, we also include ViT/S-16 models (i.e., same architecture of $f_\theta$ in Fig. 2) trained without HCS data (ViT-ImageNet) and with weak supervision of perturbation labels (ViT-WSL). Results are reported in Table 1. To estimate data and parameter efficiency together with performance, we also reported the number of training data and model parameters. Computation efficiency evaluation in terms of FLOPs are provided in Appendix A.2.

**Compound-Gene Interaction Benchmark**  A compound-gene interaction benchmark of RxRx3-core test whether a model can link gene knockouts and small-molecule perturbations by calculating the cosine similarity between their embeddings (Celik et al., 2024). For each compound, the model is evaluated on how highly it ranks known target genes over random genes, reporting area under the curve (AUC), and average precision (AP) measured based on the similarity scores. The final results are summarized as the mean and standard deviation (Std.) of AUC and AP over compounds, with comparison to a random baseline (i.e., random compound-gene relationships) via z-scores. The ground truth compound-gene relationships are curated from multiple sources, including PubChem, Guide to Pharmacology, WIPO, D3R, BindingDB, and ChEMBL (Liu et al., 2007; Zdrazil et al., 2024; Harding et al., 2024).

In Section 5.2, we have measured the performance of CellProfiler, CellCLIP, OpenPhenom, Phenom-1, Phenom-2, and two more baselines of ViT/S-16 trained with ImageNet-1K and Bray datasets. Since we didn't have access to source codes nor reported metrics of CLOOME and MolPhenix, we were unable to compare their performance in this benchmark.

### 4.3 Validation of CWA-MSN

**Single-Well vs. Cross-Well Alignment**  One of the key innovations in CWA-MSN is to utilize cross-well images as implicit data augmentation for training. In Section 5.3, we validated the effect of this cross-well sampling strategy, by changing it to a conventional single-well sampling method (i.e., $w_a^p \neq w_t^p$ for cross-well vs. $w_a^p \equiv w_t^p$ for single-well; see Section 3.2). We performed the comparison between single-well and cross-well based on the gene-gene interaction benchmark, using the Bray dataset for training.

**Masked Siamese Network vs. Masked Autoencoder**  Next, we have checked whether a masked siamese network has indeed benefits in terms of biological relationship retrieval and training efficiency compared to a popular alternative, masked autoencoder, which has been used as a backbone network for many existing methods. More precisely, we adopted a CropMAE method (Eymaël et al., 2024) which utilizes pairs of cropped images as anchor and target, but the training objective is to reconstruct a masked target image instead of the prototype alignment.

In Section 5.4, we tested CropMAE with single-well and cross-well settings, comparing their performance with that of CWA-MSN. To examine the performance and training efficiency together, we not only reported the gene-gene interaction benchmarks but also measured the training costs in the same computing environment as GPU hours (CPU: Intel Xeon Silver 4310; GPU: NVIDIA TITAN RTX; 24 GB memory). The Bray dataset was used for this experiment.

**Prototype Number Optimization**  As prototype alignment plays a key role in the training of CWA-MSN, it is important to find an optimal number of prototypes that can effectively capture biological relationships between cellular images. Therefore, we have optimized the number by changing the number of prototypes (256, 512, 1024, and 2048) and measuring the performance based on the gene-gene interaction benchmark.

## 5 RESULTS

### 5.1 GENE-GENE INTERACTION BENCHMARK

Table 1: Gene–gene interaction benchmark results of different methods. *: Values from Lu et al. (2025). **: Not publicly available. N.A.: Not available.

| Training Dataset | # Images | # Perturb. | Parameters | Method | CORUM ↑ | hu.MAP ↑ | Reactome ↑ | StringDB ↑ |
|---|---|---|---|---|---|---|---|---|
| - | - | - | - | Random | .107 | .111 | .107 | .115 |
| ImageNet-1K | 1M | - | 22M | ViT-ImageNet | .342 | .420 | .144 | .305 |
| - | - | - | - | CellProfiler | .361 | .444 | .160 | .330 |
| Bray *et al.* | 0.2M | >7K | 22M | SupCon | .242 | .271 | .123 | .224 |
| Bray *et al.* | 0.2M | >7K | 22M | ViT-WSL | .249 | .290 | .148 | .242 |
| Bray *et al.* | 0.2M | >7K | 36M | MolPhenix* | .262 | .306 | .142 | .241 |
| Bray *et al.* | 0.2M | >7K | 25M | CLOOME* | .328 | .406 | .135 | .278 |
| Bray *et al.* | 0.2M | >7K | 1,477M | CellCLIP | .354 | .416 | .145 | .307 |
| Bray *et al.* | 0.2M | >7K | 22M | SimCLR | .256 | .290 | .137 | .239 |
| RxRx3+cpg0016 | >10M | >116K | 25M | OpenPhenom | .300 | .352 | **.158** | .281 |
| RPI-93M | 93M | ~4M | 307M | Phenom-1** | .395 | .482 | .188 | .349 |
| PP-16M | 16M | N.A. | 1,860M | Phenom-2** | .486 | .553 | .197 | .415 |
| Bray *et al.* | 0.2M | >7K | 22M | **CWA-MSN (Ours)** | **.386** | **.447** | **.158** | **.327** |

As shown in Table 1, CWA-MSN outperformed all handcrafted, weakly supervised, contrastive learning methods on the benchmark gene-gene interaction, except a few large-scale private models (i.e., Phenom-1 and Phenom-2). In particular, it surpassed the SOTA weakly supervised contrastive learning method, CellCLIP, with significant performance gaps (e.g., CORUM: .354 for CellCLIP vs. .386 for CWA-MSN). Additional analysis in Appendix A.3 further verifies that our performance gains indeed stem from batch-effect mitigation. Considering that the same Bray dataset was used for CellCLIP and CWA-MSN training, these results demonstrate the superior parameter efficiency of CWA-MSN with a much smaller model size (1,477M for CellCLIP vs. 22M for CWA-MSN).

Furthermore, CWA-MSN outperformed OpenPhenom, which is publicly available SOTA self-supervised method, in most of the retrieval tasks (CORUM: .300 vs. .386, hu.MAP: .352 vs. .447, and StringDB: .281 vs. .327)). The results indicate better data efficiency of CWA-MSN compared to OpenPhenom, even with the large gap between the number of training images (>10M for Open-Phenom vs. 0.2M for CWA-MSN).

The benchmark results for Phenom-1 and Phenom-2 are in fact better than those for CWA-MSN. However, there are huge differences in model sizes and training data volumes between these methods. For example, the number of training images and model parameters of Phenom-1 is 465 times and 14 times larger than CWA-MSN, respectively (Images: 93M vs. 0.2M; Parameters: 307M vs. 22M). As summarized in Fig. 1, if we consider these aspects together, CWA-MSN has significant advantages over Phenom-1 and Phenom-2 in terms of data and parameter efficiency (e.g., see (b), (c) and (d) in Fig. 1). Also, it should be noted that the data (RPI-93M and PP-16M) and the source codes of Phenom-1 and Phenom-2 are not publicly available.

### 5.2 COMPOUND-GENE INTERACTION BENCHMARK

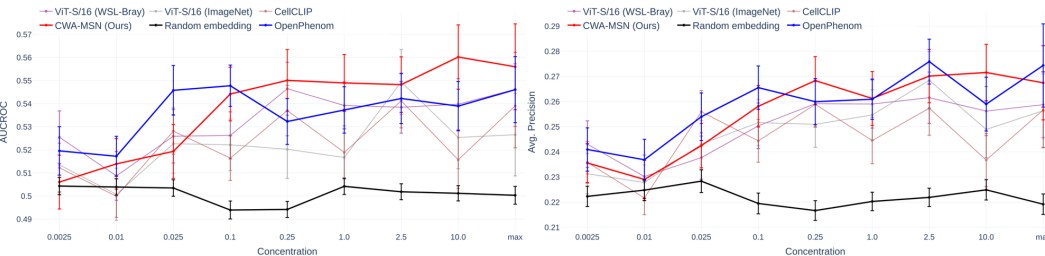

Figure 3: Compound-gene interaction benchmark graphs. AUC-ROC and AP values are reported over concentration.

Table 2: Compound-gene interaction benchmark results at the maximum concentration level. The best performance is in **bold**, and second best is in underline. *: Not publicly available. **: Evaluated using open models.

| Method | AP | | | AUC-ROC | | |
|---|---|---|---|---|---|---|
| | Mean | Std. | Z-score↑ | Mean | Std. | Z-score↑ |
| Phenom-2* | .307 | .015 | 6.04 | - | - | - |
| Phenom-1* | .290 | .017 | 4.35 | - | - | - |
| OpenPhenom** | .274 | .017 | **3.89** | .546 | .014 | **3.16** |
| ViT-WSL | .259 | .013 | 3.37 | .546 | .016 | 2.79 |
| CellProfiler | .276 | .018 | 3.34 | - | - | - |
| CellCLIP** | .257 | .015 | 2.81 | .539 | .018 | 2.12 |
| ViT-ImageNet | .256 | .015 | 2.75 | .527 | .018 | 1.46 |
| Random | .214 | .003 | 0.00 | .500 | .004 | 0.00 |
| CWA-MSN (ours) | .267 | .015 | 3.55 | .556 | .019 | 2.88 |

Fig. 3 shows the graphs of the compound-gene interaction benchmark for each method, reporting the AUC-ROC and AP values over the concentration. In general, the graphs of CWA-MSN and OpenPhenom are competing with each other as the top performing method. For example, CWA-MSN consistently outperforms all other methods in the AUC-ROC graph within a range of 0.25 $\mu$ Mol up to the maximum concentration, whereas OpenPhenom dominates in the other range of concentrations (see Fig. 3).

If we closely investigate the z-scores of each method at the maximum concentration, OpenPhenom achieved the highest z-scores in both the AP and AUC-ROC metrics (3.89 and 3.16) compared to the second-best z-scores of CWA-MSN (3.55 and 2.88) as shown in Table 2. Although the z-scores of CWA-MSN are slightly lower than those of OpenPhenom, these two methods possibly have complementary strengths. For example, the std. of AP is slightly lower in CWA-MSN (i.e., better feature consistency among known relationships) than that of OpenPhenom, whereas the mean AP is marginally higher in OpenPhenom (i.e., better capturing known relationships on average).

Most importantly, we want to highlight that this competitive performance of CWA-MSN relative to OpenPhenom, was achieved despite training in a significantly smaller data size. For example, as shown in Table 1, OpenPhenom (and also Phenom-1 and Phenom-2) requires a 50 times higher number of training images (>10M for OpenPhenom vs. 0.2M for CWA-MSN), covering 17,063 gene knockouts and 1,674 compounds in more than 180 experimental batches. Despite this massive training advantage, OpenPhenom achieved modestly higher scores than those of CWA-MSN,

### 5.3 SINGLE-WELL VS. CROSS-WELL ALIGNMENT

Table 3: Gene-gene interaction benchmark results between single-well and cross-well masked simease networks. The best performance is highlighted in **bold**.

| Model | CORUM | hu.MAP | Reactome | StringDB |
|---|---|---|---|---|
| # relationships | *1,209* | *958* | *569* | *1,737* |
| Random | .107 | .111 | .107 | .115 |
| Single-Well-MSN | .281 | .330 | .130 | .261 |
| CWA-MSN (Ours) | **.386** | **.447** | **.158** | **.327** |

As summarized in Table 3, when we tested the effect of single-well and cross-well sampling strategies combined with a masked simease network, we observed significant performance gaps between the two models. Concretely, compared to the single-well alignment (i.e., Single-Well-MSN in Table 3), the cross-well alignment (i.e., CWA-MSN in Table 3) largely improves recall in all gene-gene association databases, including CORUM (from .281 to .386), hu.MAP (from .330 to .447), Reactome (from .130 to .158), and StringDB (from .261 to .327). These findings show that cross-well sampling consistently outperforms the single-well counterpart in biological relationship retrieval.

## 5.4 Masked Siamese Network vs. Masked Autoencoder

Table 4: Gene–gene interaction benchmark comparison of CWA-MSN and CropMAE. The best performance per metric is highlighted in **bold**.

| Training Time (GPU hours) | Model | | CORUM | hu.MAP | Reactome | StringDB |
|---|---|---|---|---|---|---|
| | | # relationships | 1,209 | 958 | 569 | 1,737 |
| - | Random | | .107 | .111 | .107 | .115 |
| 109 | CropMAE-Single | | .338 | .408 | .137 | .303 |
| 14 | CropMAE-Cross | | .348 | .443 | .135 | .309 |
| <9 | CWA-MSN (Ours) | | **.386** | **.447** | **.158** | **.327** |

Table 4 shows that CWA-MSN consistently surpasses CropMAE (Eymaël et al., 2024) with either single-well or cross-well settings in gene-gene relationship retrieval tasks. In detail, compared to CropMAE with cross-well sampling (i.e., CropMAE-Cross), CWA-MSN achieved higher recall in all gene-gene interaction databases (e.g., .348 vs. .386 in CORUM) with the minimum training time (14 vs. <9 GPU hours). The results indicate that applying cross-well alignment strategy to a masked siamese network (prototype-based learning) is a more effective combination than to a masked autoencoder (reconstruction-based learning) in terms of performance and training cost. Interestingly, applying the proposed cross-well sampling strategy to CropMAE alone substantially reduced training cost (i.e., from 109 to 14 GPU hours) while also improving benchmark performance (see CropMAE-Single vs. CropMAE-Cross in Table 4).

## 5.5 Prototype Number Optimization

Table 5: Optimization results for the number of prototypes in CWA-MSN based on gene-gene interaction prediction. The best performance for each metric is highlighted in **bold**.

| Number of Prototypes | | CORUM | hu.MAP | Reactome | StringDB |
|---|---|---|---|---|---|
| | # relationships | 1,209 | 958 | 569 | 1,737 |
| 256 | | .372 | .433 | .132 | .321 |
| 512 | | .344 | .401 | .151 | .311 |
| 1,024 | | **.386** | **.447** | **.158** | **.327** |
| 2,048 | | .369 | .438 | .141 | .314 |

The optimization results for the number of prototypes is summarized in Table 5. When we changed the number from 256 to 2,048, the best performance was achieved at the number equal to 1,024. We potentially concluded that this is a point that balances the redundancy and diversity of prototypes.

## 6 Conclusion

In conclusion, we present CWA-MSN, a simple and effective framework for representation learning of cell painting images, which can extract phenotypic changes according to chemical and genetic perturbations with high data and parameter efficiency. By aligning embeddings of identically perturbed cells across wells using a masked siamese architecture, CWA-MSN mitigates batch effects while preserving fine-grained morphology. This yields biologically meaningful features that improve relationship retrieval across gene–gene and compound–gene, surpassing state-of-the-art public self-supervised and contrastive baselines, even under limited data and parameter budgets.

## 7 Disclosure of LLM usage

We used large language models (ChatGPT and Claude) to assist with code design and manuscript editing. All outputs were reviewed and validated by the authors, who take full responsibility for the accuracy and originality of this work.

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

# A APPENDIX

This supplement presents extended methodological details, ablations, and quantitative analyzes in support of the main text. These experiments provide further evidence for the contribution of the masking mechanism(A.1), computational efficiency(A.2), reduction of batch effects(A.3), and distinctions between CWA-MSN and conventional natural-image SSL adaptations(A.4).

## A.1 CONTRIBUTION OF THE MASKING MECHANISM

CWA-MSN is trained with weak supervision from perturbation labels, which raises the natural question of whether the masking mechanism still contributes meaningfully in this setting. To assess its role, we performed an ablation study that isolates the effect of the asymmetric masking design.

**Ablation: Masking vs. No Masking.** We compare our masked-reconstruction scheme against a no-masking variant. The two training settings are identical except for the masking rate, which is set to zero for the no-masking model. Table 6 reports the performance of the RxRx3-core gene–gene interaction benchmark.

Table 6: Ablation comparing CWA-MSN with the no-masking variant.

| Model | CORUM | hu.MAP | Reactome | StringDB |
|---|---|---|---|---|
| **CWA-MSN (Ours)** | **0.386** | **0.447** | **0.158** | **0.327** |
| No Masking | 0.354 | 0.423 | 0.147 | 0.320 |
| CellCLIP | 0.354 | 0.416 | 0.145 | 0.307 |
| OpenPhenom | 0.300 | 0.352 | 0.158 | 0.281 |

The performance gain of our method compare to method without masking indicates that asymmetric masking provides a clear benefit even in the presence of perturbation supervision. By masking only the anchor view, the model is forced to rely on perturbation-relevant morphological cues rather than low-level artifacts, improving robustness under the noisy and batch-variable conditions of cell-painting data.

## A.2 COMPUTATIONAL EFFICIENCY (FLOPs ANALYSIS)

In the main text, we show that CWA-MSN matches or surpasses prior methods under constrained data and parameter budgets, indicating better data and parameter efficiency. To further assess computational efficiency, we compare training FLOPs.

Table 7: Model complexity and gene–gene interaction retrieval performance on RxRx3-core.

| Method | GFLOPs | #Params (M) | CORUM | hu.MAP | Reactome | StringDB |
|---|---|---|---|---|---|---|
| ViT-WSL | 8.79 | 22 | 0.249 | 0.290 | 0.148 | 0.242 |
| CellCLIP | 339.17 | 1,477 | 0.354 | 0.416 | 0.145 | 0.307 |
| OpenPhenom | 104.68 | 25 | 0.300 | 0.352 | 0.158 | 0.281 |
| **CWA-MSN (Ours)** | **23.66** | **22** | **0.386** | **0.447** | **0.158** | **0.327** |

**Interpretation.** CWA-MSN achieves state-of-the-art biological performance while maintaining a computation footprint far below large-scale alternatives. These results show the consistently superior efficiency of our method in all aspects.

## A.3 PROBING BATCH-EFFECTS

This section provides quantitative evidence that CWA-MSN reduces batch effects compared to current publicly available baselines.

**Evaluation Protocol.** Batch effects in image-based profiling are commonly quantified by measuring how well technical metadata (plate, batch, acquisition day) can be recovered from embeddings. High predictability indicates substantial non-biological variation, whereas low predictability signals successful batch-effect suppression. We follow this standard practice by probing the recoverability of plate identity via linear classifiers and $k$NN ($k$=5) with 5-fold cross-validation. We evaluate on the full RxRx3-core dataset as well as a variant with all negative controls removed.

Table 8: Five-fold cross-validation macro-F1 for predicting plate identity from learned embeddings.

| Embedding | Linear | | KNN | |
| | Full↓ | No Ctrl↓ | Full↓ | No Ctrl↓ |
| --- | --- | --- | --- | --- |
| CWA-MSN | 13.22% ± 0.64% | 13.99% ± 0.85% | 13.32% ± 0.22% | 13.34% ± 0.33% |
| OpenPhenom | 27.07% ± 0.55% | 28.32% ± 0.37% | 26.83% ± 0.15% | 27.23% ± 0.46% |

**Interpretation.** Across all probing strategies, CWA-MSN yields significantly lower plate-predictability (approximately half) than OpenPhenom, indicating substantially weaker entanglement with batch-specific artifacts. These results corroborate the cross-well alignment strategy described in the main text and demonstrate that the method effectively suppresses technical variation while preserving morphological signal.

A.4   HCS MODAL-SPECIFIC DESIGNS

This section elaborates on the methodological distinctions between CWA-MSN and conventional SSL frameworks originally developed for natural images.

**Choice of MSN Backbone.** We select MSN over alternatives such as DINOv2 because its prototype-aligned masked-Siamese objective offers a stable learning signal with minimal augmentation requirements, which is essential in microscopy where strong natural-image augmentations such as Gaussian Blur can distort subtle perturbation-dependent features.

**Lightweight and Data-Efficient Training.** Incorporating MSN's masking mechanism enables efficient training by removing the need for self-reconstruction (MAE) or contrastive sampling, which significantly reduces computational overhead and fits the constraints of modeling perturbation effects under limited supervision and realistic compute budgets. Thus, Unlike prior work such as Phenom-1/2, which relies on ultra large datasets and models, our lightweight architecture (22M parameters, 0.2M images) along with critical sampling strategy intentionally ease the learning of perturbation effects.

**Cross-Well Alignment** Most importantly, CWA-MSN introduces the cross-well alignment strategy, a domain-specific contribution that forces representations from wells sharing the same perturbation to align across plates, directly mitigating batch and well-specific variation while preserving true morphological signal

Together, these design choices make CWA-MSN a purpose-built approach for morphological profiling rather than a straightforward adaptation of MSN to a new domain.

