# OpenReview forum: "Efficient Cell Painting Image Representation Learning via Cross-Well Aligned Masked Siamese Network"
_ICLR.cc/2026/Conference — Submitted to ICLR 2026_

### Official Review · Reviewer_aUNZ · 2025-10-27

**Soundness:** 3
**Presentation:** 3
**Contribution:** 2
**Rating:** 4
**Confidence:** 5

**Summary:**

This paper proposes a self-supervised contrastive learning framework for representation learning on Cell Painting images. The proposed approach achieves strong data efficiency and outperforms existing methods.

**Strengths:**

The paper addresses an important and timely problem in morphological profiling by proposing an efficient supervised contrastive learning framework for Cell Painting images. The proposed cross-well sampling strategy, which leverages plate and batch diversity as a form of data augmentation, is an intuitive and biologically grounded idea that improves robustness to batch effects. The method demonstrates strong performance in data-efficient regimes and shows potential for scalable representation learning in high-content imaging in RxRx3-core.

**Weaknesses:**

**Motivation & technical contribution**
* While the proposed cross-well sampling strategy across batches and plates is an interesting data augmentation approach for Cell Painting data, the overall proposed method relies heavily on existing architectures and previously established contrastive SSL, limiting methodological novelty.

* The authors argue that their approach provides a stronger proxy training signal for capturing perturbation effects than prior methods. However, since supervision is based primarily on the same pairs, it remains unclear why this setup would effectively distinguish true perturbation effects from false positives, perturbations that are visually similar to controls.

**Limited evaluation**
* The evaluation is relatively narrow in scope. Given that the proposed method is generalized for Cell Painting images, experiments beyond RxRx3-Core would strengthen the work. For instance, evaluating on CP-JUMP1 [1] or reporting performance on the Bray et al. dataset would better demonstrate generalizability.

* Since the method is presented as a contrastive self-supervised learning framework, it would be important to include baselines that leverage supervision in comparable ways (e.g., standard supervised or semi-supervised contrastive methods, or some other augmentation strategies [2,3]). Such comparisons would provide a fairer and more comprehensive evaluation of the proposed approach.

* Similarly, the paper does not provide sufficient detail or analysis on how the design choices, such as the criteria for selecting anchors and target pairs, affect the effectiveness of the proposed method. Since the formulation primarily emphasizes positive-pair similarity without any negative pairs, it remains uncertain whether the method can effectively handle one of the key downstream tasks, sister perturbation identification [1], where morphologically similar but biologically distinct perturbations must be distinguished.

* To support the claims of computational efficiency, the authors should report the number of FLOPs, as this is a more standard and informative metric for evaluating efficiency than parameter count alone.

[1] Three million images and morphological profiles of cells treated with matched chemical and genetic perturbations. Nature com, Chandrasekaran et al.

[2] Supervised Contrastive Learning. NeurIPS 2020, Khosla et al.

[3] SimCLR (Self/Semi-supervised Baseline) ICML 2020, Chen et al.

**Questions:**

1. Are there alternative criteria or strategies for selecting anchors and target pairs in practice, and how sensitive is the model’s performance to these design choices? How does the number of positive pairs affect the model's effectiveness?
2. Since SSL encourages similarity, to what extent does this objective reduce or obscure meaningful individual variance in the learned representations?
3. Could there be scenarios where pairing samples across plates might inadvertently reinforce batch effects rather than mitigate them?
4. Why does the CropMAE-single require longer training time compared to CropMAE-cross?

---

> ### Author Response · Authors · 2025-11-21
> **[Part 1] Weakness 1-4**
>
> We sincerely thank the reviewer for highlighting the biological intuition and robustness of our cross-well sampling strategy and its strong performance in data-efficient regimes.
>
> We address the raised concerns below.
>
> # Weakness
>
> > **1. "Limited methodological novelty"**
>
> We highlight the novelty of CWA-MSN as follows:
>
> Instead of naive application of Masked Siamese Network to Cell Painting data, CWA-MSN integrates design choices specific to morphological profiling, including a stable prototype-aligned objective with minimal augmentations, a lightweight architecture, and most importantly a cross-well sampling and alignment strategy that mitigates batch effects while preserving true biological signal.
>
> Detailed explanations for each component are provided in the global comment above **(CLARIFICATION-3)**.
>
> > **2. "Positive pair only scheme may hinder the sister perturbation retrieval (Weakness 2, Weakness 5-2)"**
>
> **Weakness 2** concerns the ability to distinguish perturbations visually similar to controls, while **Weakness 5-2** questions the ability to distinguish between morphologically similar but biologically distinct perturbations.
>
> Both issues relate to maintaining discriminative structure in a positive-pair–only framework.
>
> **1. Architectures uses only positive pairs are well-established**
>
> Prior work such as BYOL and SimSiam demonstrates that architectural asymmetry and stop-gradient are sufficient to avoid collapse and to learn separable representations even among visually similar classes. CWA-MSN shares these mechanisms.
>
> **2. The ME-MAX regularizer**
>
> The ME-MAX regularizer encourages diverse prototype usage, reducing the chance that distinct perturbations are mapped to the same representation unless truly indistinguishable. This supports both non-collapse against controls and separation among similar perturbations.
>
> **3. Empirical evidence**
>
> CWA-MSN achieves substantial gains on gene–gene and compound–gene retrieval benchmarks, tasks that require capturing subtle and biologically meaningful perturbation differences. These improvements indicate that the model reliably separates both weak perturbations from controls and distinct perturbations from each other.
>
> > **3. "Narrow evaluation scope, limited to RxRx3-core"**
>
> We agree that broader evaluations are valuable. However, gene–gene and compound–gene retrieval remain among the most biologically interpretable tests for perturbation-aware representations.
>
> High-quality Cell Painting benchmarks are scarce, with many datasets either extremely large (e.g., CP-JUMP1, RxRx3) or private (e.g., RPI-93M, PP-16M) and lacking the biologically meaningful evaluations we use.
>
> In contrast, **RxRx3-core** is carefully curated, experimentally controlled, and paired with reliable perturbation ground truth, making it a suitable benchmark for assessing whether a model captures true biological effects. While we hope to explore additional tasks such as MoA prediction and cross-dataset transfer in future work, we believe the current results already provide meaningful evidence of CWA-MSN’s effectiveness.
>
> > **4. "Evaluation is limited to _cell painting_ SSL"**
>
> We note that a conventional supervised baseline is already included in our paper as ViT-WSL, following the setup of [1].
>
> A more direct supervision, e.g., contrastive learning based on interaction labels, is challenging because such labels are relatively scarce.
>
> While other contrastive-based or semi-supervised methods could serve as baselines, they inherently rely on tailored, often strong augmentations to generate sufficient diversity, which can distort the subtle morphological cues induced by perturbations.
>
> To illustrate this, we provide a comparison with SupCon and SimCLR, demonstrating that CWA-MSN outperforms augmentation-driven contrastive methods while remaining augmentation-light and better aligned with the biological signal.
>
> **RxRx3-core Gene-gene Interaction Benchmark Performances of natural-image SSL methods**
>
> | Method          | CORUM | hu.MAP | Reactome | StringDB |
> |-----------------|-------|--------|----------|----------|
> | SupCon          | 0.242 | 0.271  | 0.123    | 0.224    |
> | SimCLR          | 0.256 | 0.290  | 0.137    | 0.239    |
> | CWA-MSN (Ours)  | 0.386 | 0.447  | 0.158    | 0.327    |
>
> [1] Moshkov et al. *Learning representations for image-based profiling of perturbations. Nat. Commun.* (2024)

---

> ### Author Response · Authors · 2025-11-21
> **[Part 2] Weakness 5-6, Questions 1-4**
>
> > **5. "Insufficient design choice detail (Weakness 5-1 Question 1)"**
> >
> > - **a. Sensitivity of model to design choices?**
> > - **b. Effect of number of positive pairs to the model's effectiveness?**
>
> We address concerns about critical design choices in the following:
>
> **1. Limited design choices for labels:**
>
> Current datasets have limited institute, machine, or technical metadata restricts strategies like cross-institute/machine positive pair alignment.
>
> **2. Experiment to explore design choices:**
>
> In Section 5.3 of our paper, we compared single-well cross-site alignment with our Cross-Well Alignment strategy.
> Results show substantial gains from cross-well alignment, underscoring the model’s sensitivity to design choices and the need to capture true perturbation effects rather than site positions.
>
> **3. Number of positive pairs:**
>
> We also performed additional experiments by varying the number of anchor wells (i.e., positive pairs) while keeping the total number of views fixed in the following table.
> Results show no clear trend, suggesting more complex mechanisms govern the number of anchors aligned simultaneously.
>
> We hypothesize that including more wells may introduce conflicting signals between true perturbation and batch effects, potentially confusing the model.
>
> The current setting achieves the best performance with a simpler sampling strategy.
>
> **Effect of Number of Positive Pairs**
>
> | # Positive Pairs | CORUM | hu.MAP | Reactome | StringDB |
> | --- | --- | --- | --- | --- |
> | 1 (Ours) | 0.386 | 0.447 | 0.158 | 0.327 |
> | 2 | 0.326 | 0.416 | 0.134 | 0.309 |
> | 4 | 0.368 | 0.426 | 0.142 | 0.324 |
> | 8 | 0.327 | 0.380 | 0.139 | 0.296 |
>
> > **6. "Report FLOPS along with parameter count"**
>
> We agree that FLOPs alongside parameters better reflect computational efficiency.
> However, in our context, "efficiency" refers to achieving similar or better performance under constrained training data and parameter budgets.
> We report training FLOPs below, consistently showing our method’s superior efficiency.
>
> **Model Complexity with RxRx3-core Gene-gene Interaction Benchmark Performances**
>
> | Method | # GFLOPs | Params | CORUM | hu.MAP | Reactome | StringDB |
> | --- | --- | --- | --- | --- | --- | --- |
> | ViT-WSL | 8.79 | 22M | 0.249 | 0.290 | 0.148 | 0.242 |
> | CellCLIP | 339.17 | 1,477M | 0.354 | 0.416 | 0.145 | 0.307 |
> | OpenPhenom | 104.68 | 25M | 0.300 | 0.352 | 0.158 | 0.281 |
> | Ours (CWA-MSN) | 23.66 | 22M | 0.386 | 0.447 | 0.158 | 0.327 |
>
> # Questions
>
> > **1. "How much does this objective obscure meaningful individual variance in the learned representations?"**
>
> CWA-MSN does not collapse meaningful within-perturbation variability.
>
> CWA-MSN differs from standard self-supervision, combining:
>
> - Masked MSN-style reconstruction
>
> - Prototype-based assignment with entropy regularization
>
> - Weak supervision via perturbation labels
>
> These encourage semantic alignment across wells while preserving representational richness through masking and prototype diversity.
>
> - Representation Goal
>
> The variance to preserve depends on high-content screening objectives. For RxRx3-core, the aim is to maximize perturbation-level biological signal while minimizing nuisance variation (batch, plate, imaging noise). Reducing uncontrolled individual-level variation is desirable if meaningful distinctions are retained.
>
> - Empirical Evidence
>
> CWA-MSN preserves these distinctions, outperforming OpenPhenom and CellCLIP on gene–gene retrieval and other tasks requiring fine-grained morphology, showing alignment focuses on perturbation-relevant biology without obscuring meaningful variability.
>
> - Cell-level Variance
>
> Analyzing cell-level variance retention is an interesting extension and a promising direction for future work.
>
> > **2. "Can cross-plate pairing unintentionally amplify batch effects?"**
>
> Cross-plate pairing could theoretically reinforce batch effects if plate-specific biases such as well position are incorporated.
>
> CWA-MSN avoids this risk in two ways:
>
> - alignment is performed only between wells with the same perturbation, using the biological signal as the anchor;
>
> - the masked Siamese objective encourages the model to focus on invariant morphological features rather than plate-specific noise.
>
> Consistently, our experiments show better recovery of true perturbation relationships, indicating the alignment reinforces biologically meaningful signals over technical artifacts.
>
> > **3. "Why does the CropMAE-single trains longer than CropMAE-cross?"**
>
> CropMAE-single trains the model to reconstruct each well’s masked image, so the dataset length equals the total number of wells.
>
> CropMAE-cross reconstructs an image using another well from the same perturbation, operating at the perturbation level. This makes it a fairer comparison to our design, with dataset length equal to the number of perturbations.
>
> As a result, CropMAE-single has significantly longer training time, since each perturbation contains multiple wells.

---

### Official Review · Reviewer_sW6k · 2025-10-31

**Soundness:** 3
**Presentation:** 3
**Contribution:** 2
**Rating:** 4
**Confidence:** 3

**Summary:**

In this paper, the authors introduce the Cross-Well Aligned Masked Siamese Network (CWA-MSN), a strategy designed to align HCS image embeddings of identically perturbed cells across wells. The proposed framework outperforms existing methods in gene–gene and compound–gene retrieval, even when trained with limited data.

**Strengths:**

1. The paper addresses an interesting problem in HCS imaging, where biological information can be obscured by batch effects.
2. The paper is well written and easy to follow.
3. The proposed framework achieves promising results in gene–gene and compound–gene retrieval.

**Weaknesses:**

1. The novelty of the work is somewhat limited, as it can be viewed as an application of (Assran et al.) to HCS images.

2. I think that the evaluation tasks used (gene–gene and compound–gene retrieval) are rather limited. Did the authors apply the trained model to any other tasks?

**Questions:**

1. Did the authors quantify the batch effect? Does the method provide improvements in that regard?

2. Did the authors train on larger datasets? Would that be expected to further improve the results?

---

> ### Author Response · Authors · 2025-11-21
>
> We appriciate the reviewer for recognizing the clarity of our presentation and the promising results of our framework.
>
> We address the concerns raised in detail below.
>
> > **1. "Limited Novelty of application of Masked Siamese Network to HCS images**
>
> We emphasize the novelty of CWA-MSN as follows:
>
> Rather than directly applying MSN to Cell Painting data, it incorporates design choices tailored to morphological profiling: a stable prototype-aligned objective with minimal augmentations, a lightweight architecture for small datasets, and most importantly cross-well sampling and alignment strategy that mitigates batch effects while preserving true biological signal.
>
> Detailed explanations for each aspect are provided in the global comments above **(CLARIFICATION-3)**.
>
> > **2. "Limited evaluation tasks applied besides gene–gene and compound–gene retrieval"**
>
> Broader evaluations are valuable; however, gene–gene and compound–gene retrieval remain among the *most biologically rigorous and interpretable* tests for perturbation-aware representation learning. High-quality benchmarks for cell-painting data are extremely limited, whereas most existing cell painting datasets are either extremely huge (e.g., CP-JUMP1, RxRx3) or private (e.g., RPI-93M, PP-16M) and lack the specific downstream evaluations used in the paper.
>
> In contrast,  **RxRx3-core** is one of the few datasets that is carefully curated, experimentally controlled, and paired with reliable perturbation ground truth, making it an strong benchmark for assessing whether a model captures true biological effects.
>
> While we plan to explore additional tasks (e.g., MoA prediction, dose–response modeling, cross-dataset transfer) in future work, we believe that the current results already provide strong evidence that CWA-MSN learns efficient and biologically meaningful embeddings.
>
> > **3. "Improvement regarding quantified batch effect"**
>
> We have conducted an evaluation to quantify the batch effects.
>
> This analysis aligns with established practice in image-based profiling, where predicting technical attributes such as plate or batch labels is widely used to quantify non-biological variation [1]. In these frameworks, higher predictability indicates stronger batch effects because the representation retains more technical—rather than biological—signal.
>
> **Quantifying Batch Effects.**
>
> We evaluate the recoverability of plate identifiers from each embedding using linear and KNN (k=5) probing with 5-fold cross-validation. Plate identity is broadly treated as a practical surrogate for technical variation; accordingly, probing performance offers an informative indication of batch-effect severity.
>
> We compare our model (CWA-MSN) with OpenPhenom. Experiments were run on the full RxRx3-core dataset and a version excluding negative controls. Across all settings, CWA-MSN yields the lowest plate-prediction macro-F1 (≈13%), well below OpenPhenom (≈26–28%) with low variance.
>
> **5-Fold CV Probing Performance on "Plate" (Macro-F1)**
>
> | Embedding   | Linear (Full)        | Linear (No Ctrl)      | KNN (Full)            | KNN (No Ctrl)         |
> |-------------|-----------------------|------------------------|------------------------|------------------------|
> | CWA-MSN     | 13.22% ± 0.64%        | 13.99% ± 0.85%         | 13.32% ± 0.22%         | 13.34% ± 0.33%         |
> | OpenPhenom  | 27.07% ± 0.55%        | 28.32% ± 0.37%         | 26.83% ± 0.15%         | 27.23% ± 0.46%         |
>
> **Interpretation.**
>
> These additional results provide consistent quantitative evidence that CWA-MSN encodes less batch effect than existing baselines.
>
> [1] Arevalo et al. *Evaluating batch correction methods for image-based cell profiling. Nat. Commun.* (2024)
>
> > **4. "Expected effect of larger training datasets"**
>
> We did not train on larger datasets as most existing Cell Painting datasets are either extremely large (e.g., CP-JUMP1 → 700 TB, RxRx3 → 83 TB) or private (e.g., RPI-93M, PP-16M), and some lack the corresponding modalities (CRISPR, ORF) required for our downstream evaluation.
>
> Nonetheless, as the reviewer pointed out, prior work (OpenPhenom) has shown that scaling data improves performance (e.g., RPI-52M → RPI-93M), suggesting that larger datasets would likely further benefit CWA-MSN.
>
> To explore this effect in a tractable setting, we conducted controlled experiments by reducing the number of perturbations (i.e., reducing data-size) in Bray dataset we used.
>
> The result illustrate that increasing the number of perturbations improves representation quality, consistent with the expected benefits of larger datasets.
>
> **Performance vs. Number of Perturbations**
>
> | # Perturbations | CORUM | hu.MAP | Reactome | StringDB |
> |-----------------|-------|--------|----------|-----------|
> | 10%             | 0.352 | 0.431  | 0.130    | 0.312     |
> | 50%             | 0.361 | 0.430  | 0.141    | 0.324     |
> | 100% (Ours)     | 0.386 | 0.447  | 0.158    | 0.327     |

---

### Official Review · Reviewer_MPNE · 2025-11-01

**Soundness:** 2
**Presentation:** 2
**Contribution:** 2
**Rating:** 4
**Confidence:** 5

**Summary:**

This paper introduces the Cross-Well Aligned Masked Siamese Network (CWA-MSN), designed to address challenges in extracting robust biological features from high-content screening images. Specifically, CWA-MSN targets the issue of batch effects, such as variations in experimental conditions. The framework aligns cell representations from different wells subjected to the same perturbation, ensuring consistency in the learned features despite these batch effects.

**Strengths:**

Aligning cell representations from different wells within the same perturbation is a reasonable approach.

The method shows improvements in performance with fewer data and model parameters compared to existing methods.

**Weaknesses:**

The idea of aligning representations under the same perturbation has already been proposed in [1]. Additionally, [1] also uses a self-distillation framework with a masked modeling approach. While there are differences in design, this paper employs the MSN framework, while [1] uses DINO v2, and this paper aligns images from different wells, whereas [1] constructs positive pairs from different sites within the same well. A thorough discussion is needed to clarify the unique contributions of this paper.

Another concern relates to the problem setup. While this paper uses the perturbation label to determine the target for cross-well alignment, this approach no longer adheres to a self-supervised setting. As a result, the claim of addressing batch effects in self-supervised approaches seems questionable. Additionally, this claim is not supported by experiments or visualizations.

[1] Self-supervised Representation Learning with Local Aggregation for Image-based Profiling,	arXiv:2506.14265.

**Questions:**

Given the distributional gap between cell images and natural images, how do you handle the different input channels and augmentations?

---

> ### Author Response · Authors · 2025-11-21
>
> We thank the reviewer for positive remarks, noting the value of aligning cell representations across wells and the method’s strong performance with fewer data and parameters.
>
> We address all concerns in the following.
>
> > **1. "Thorough discussion is needed to clarify the unique contributions of this paper with respect to SSLProfiler [1]"**
>
> To clarify our contributions relative to SSLProfiler [1], we emphasize three distinctions.
>
> First, CWA-MSN adopts a simpler two-branch masked-siamese architecture with prototype-based objective, whereas SSLProfiler relies on DINOv2 with multiple auxiliary losses and site-level aggregation.
>
> Second, our method performs explicit cross-well sampling and alignment, targeting the cross-well and cross-plate batch effects rather than within-well variation of [1].
>
> Third, by focusing on perturbation-consistent signals across experimental contexts, CWA-MSN provides a more efficient and biologically grounded representation learning.
>
> We detailed the clarifications in the corresponding global comment on the top **(CLARIFICATION-2**).
>
> [1] Dai et al., *Self-supervised Representation Learning with Local Aggregation for Image-based Profiling - CVPR CVDD* (2025)
>
> > **2. "The claim of addressing batch effects in SSL approach is questionable."**
>
> CWA-MSN is indeed a weakly supervised method, leveraging perturbation labels to guide cross-well alignment.
>
> We include both weakly supervised baselines (CellCLIP, ViT-WSL, MolPhenix, CLOOME) and self-supervised baselines (OpenPhenom, Phenom-1/2) to ensure a comprehensive evaluation.
>
> Importantly, CWA-MSN achieves strong performance with higher data and parameter efficiency, even compared to weakly supervised methods.
>
> We give detailed explanation in the corresponding global comment on top **(CLARIFICATION-1)**.
>
> > **3. "The claim of addressing batch effects in self-supervised approaches is not supported by experiments or visualizations."**
>
> As noted in global comment **(CLARIFICATION-1)**, CWA-MSN is proposed under the setting of weakly-supervised representation learning. To support with experiments, we have conducted an additional evaluation.
>
> We compared CWA-MSN to OpenPhenom by training linear and KNN (k=5) probes to predict plate identifiers using 5-fold cross-validation. The evaluation was performed on both the RxRx3-core dataset and a version excluding negative controls.
>
> **5-Fold CV Probing Performance on "Plate" (Macro-F1)**
>
> | Embedding   | Linear (Full)        | Linear (No Ctrl)      | KNN (Full)            | KNN (No Ctrl)         |
> |-------------|-----------------------|------------------------|------------------------|------------------------|
> | CWA-MSN     | 13.22% ± 0.64%        | 13.99% ± 0.85%         | 13.32% ± 0.22%         | 13.34% ± 0.33%         |
> | OpenPhenom  | 27.07% ± 0.55%        | 28.32% ± 0.37%         | 26.83% ± 0.15%         | 27.23% ± 0.46%         |
>
> Across all settings, CWA-MSN achieves **substantially lower** plate-prediction macro-F1 (≈13%) than OpenPhenom (≈26–28%), with consistently low variance.
>
> These results provide direct experimental support that our method reduces plate-associated non-biological variation more effectively than a state-of-the-art baseline.
>
> > **4. "Handle different input channels and augmentations with the distributional gap between cell images and natural images"**
>
> Our design focuses on efficient cell-image representation learning while reducing batch effects that obscure perturbation signals.
>
> Instead of adopting channel-specific tokenization approaches such as CA-MAE [1] or ChannelViT [2], which add notable computational overhead, we follow Phenom2 [3] showing that a standard ViT with proper scaling performs competitively while using fewer tokens.
>
> Accordingly, we simply adapt the ViT-S input convolution to accept 6-channel Cell Painting images.
>
> For augmentations, as noted in Section 2.1 in the paper, DINO relies on sophisticated natural-image-specific transformations to create contrastive views, whereas MSN uses prototype alignment and does not require heavy augmentations.
>
> Following [4], we use only lightweight augmentations including random resized crop and horizontal flip and avoid transformations like Gaussian Blur that may distort subtle morphological features.
>
> This streamlined design enables MSN to integrate effectively with Cell Painting data while maintaining computational efficiency.
>
> [1] Kraus et al., *Masked Autoencoders for Microscopy are Scalable Learners of Cellular Biology, CVPR* (2024).
>
> [2] Bao et al., *Channel Vision Transformers: An Image is Worth 1×16×16 Words, ICLR* (2024).
>
> [3] Kenyon-Dean et al., *ViTally Consistent: Scaling Biological Representation Learning for Cell Microscopy, ICML* (2025).
>
> [4] Bendidi et al., *Exploring Self-Supervised Learning Biases for Microscopy Image Representation, Biological Imaging* (2024).

---

### Official Review · Reviewer_WnET · 2025-11-03

**Soundness:** 2
**Presentation:** 3
**Contribution:** 2
**Rating:** 4
**Confidence:** 4

**Summary:**

This paper proposes the Cross-Well Aligned Masked Siamese Network (CWA-MSN) to address batch effects in high-content cellular imaging. The method aligns representations of cells from different wells under the same perturbation, aiming to improve robustness to experimental variations. CWA-MSN combines perturbation-guided representation alignment with masked image reconstruction to learn stable and discriminative embeddings.

**Strengths:**

1. The paper focuses on an important and practical problem in biological image analysis — mitigating batch effects in cell-level feature learning.
2. The proposed cross-well alignment is a reasonable idea in the context of perturbation-based experiments.
3. The method achieves consistent performance gains while maintaining moderate model complexity.
4. The paper is clearly written and well-structured, making it easy to follow.

**Weaknesses:**

1. The problem and solution seem misaligned. The stated goal is to overcome batch effects (common patterns across different wells within the same batch), but the proposed method aligns different wells under the same perturbation. Therefore, the performance gain may not necessarily result from eliminating batch effects. More evidence is needed to validate that batch-effect reduction is indeed achieved.

2. The method uses perturbation labels as alignment supervision. Since perturbation type is often correlated with downstream task labels, this raises concerns about fairness when comparing with self-supervised baselines that do not use label information.

3. The necessity of masked reconstruction under this setting is unclear.  Masked modeling is effective for self-supervised representation learning; however, once perturbation labels are introduced, its contribution should be reevaluated.  A comparison between masked and purely supervised formulations would help justify the design choice.

4. The idea of aligning representations under the same perturbation has been explored in "Self-supervised Representation Learning with Local Aggregation for Image-based Profiling" (arXiv:2506.14265). The paper should discuss the relationship and distinction between CWA-MSN and this prior work, particularly regarding supervision signals and architectural differences.

**Questions:**

Please see the weaknesses.

---

> ### Author Response · Authors · 2025-11-21
>
> We thank the reviewer for the positive comments on the relevance of our cross-well alignment idea, the consistent performance gains.
>
> In the following, we address all concerns and questions in detail.
>
> > **1. “The performance gain may not necessarily result from eliminating batch effects.”**
>
> _Clarification of the definition of “batch effects”_
>
> We use “batch effects” to refer broadly to non-biological variation, consistent with standard definitions in image-based profiling and genomics such as:
>
> - Leek et al. [1]: “Batch effects are sub-groups of measurements that have qualitatively different behavior across conditions and are unrelated to the biological or scientific variables in a study.”
> - Arevalo et al. [2]: “Batch effects are data variations that are not due to the biological variables being studied, but rather due to unintended technical differences.”
>
> Under this definition, aligning wells under the same perturbation is intended to suppress non-biological sources of variation that obscure biological signal, rather than to address only within-batch spatial variation (which the reviewer emphasized). We will clarify this definition in the revision.
>
> _Evidence of batch-effect reduction_
>
> To directly test whether our representation encodes less non-biological variation, we conduct a *proxy task*: predicting the “plate” identifier (a technical attribute). We evaluate linear and KNN (k=5) probing with 5-fold cross-validation on embeddings from our model (CWA-MSN) and OpenPhenom. We report macro-F1 scores on the RxRx3-core dataset under both “full dataset” and “excluded negative controls” conditions.
>
> CWA-MSN achieves ~13% plate-prediction accuracy vs 26–28% for OpenPhenom, showing reduced non-biological variation. This indicates that CWA-MSN encodes significantly less plate-specific information, supporting our claim that it reduces non-biological variation.
>
> **5-Fold CV Probing Performance on "Plate" (Macro-F1)**
> | Method      | Linear (Full) | Linear (No Ctrl) | KNN (Full) | KNN (No Ctrl) |
> |------------|---------------|----------------|------------|---------------|
> | CWA-MSN    | 13.22% ± 0.64% | 13.99% ± 0.85% | 13.32% ± 0.22% | 13.34% ± 0.33% |
> | OpenPhenom | 27.07% ± 0.55% | 28.32% ± 0.37% | 26.83% ± 0.15% | 27.23% ± 0.46% |
>
> [1] Leek et al. *Tackling the widespread and critical impact of batch effects in high-throughput data. Nat. Rev. Genet.* (2010).
>
> [2] Arevalo et al. *Evaluating batch correction methods for image-based cell profiling. Nat. Commun.* (2024).
>
> > **2. “Fairness when comparing with self-supervised baselines that do not use label information.”**
>
> Thank you for raising this concern. We agree that CWA-MSN is best understood as a weakly supervised method, since it uses perturbation labels to guide cross-well alignment. We therefore compare against both SSL and supervised baselines (including ViT-WSL) to provide a comprehensive evaluation.
>
> We offer detailed clarifications in the corresponding global comments on top **(CLARIFICATION-1)**.
>
> > **3. “The necessity of masked reconstruction label is unclear if perturbation label is introduced.”**
>
> Although perturbation labels provide supervision, *asymmetric masking* remains essential. By masking only the anchor view, the model is forced to rely on perturbation-relevant morphological cues rather than low-level artifacts, improving robustness under the noisy and batch-variable conditions of cell-painting data.
>
> We conduct an ablation comparing our masked-reconstruction scheme with a no-masking variant using the gene-gene interaction benchmark.
>
> | Source          | CORUM | HuMAP | Reactome | StringDB |
> |-----------------|-------|-------|----------|----------|
> | CWA-MSN (Ours)  | 0.386 | 0.447 | 0.158    | 0.327    |
> | No-masking      | 0.354 | 0.423 | 0.147    | 0.320    |
> | CellCLIP        | 0.354 | 0.416 | 0.145    | 0.307    |
> | OpenPhenom      | 0.300 | 0.352 | 0.158    | 0.281    |
>
> The no-masking variant underperforms across all benchmarks, confirming the benefit of asymmetric masking.
>
> > **4. “Relationship, distinction, supervision signals and architectural differences with SSLProfiler [1]”**
>
> CWA-MSN uses a simpler two-branch masked-siamese design, explicit cross-well sampling, and a single prototype-based objective, whereas [1] relies on DINOv2 with multiple auxiliary losses aimed at within-well variation.
>
> Our method instead targets the dominant cross-well/plate batch effects and aligns perturbation-consistent signals more directly and efficiently.
>
> We provide careful clarifications in the corresponding individual comments on top **(CLARIFICATION-2)**.
>
> [1] Dai et al., *Self-supervised Representation Learning with Local Aggregation for Image-based Profiling - CVPR CVDD* (2025)

---

> > ### Comment · Reviewer_WnET · 2025-11-24
> >
> > Thanks for your detailed responses. The primary concerns are well-addressed.

---

> > > ### Author Response · Authors · 2025-11-28
> > >
> > > Thank you for your time and for acknowledging our response. We are encouraged that the detailed experiments successfully addressed your concerns.
> > >
> > > We agree that these results add significant value to the paper. We will ensure that the material presented in our rebuttal is fully integrated into the final version.

---

### Author Response · Authors · 2025-11-21
**Common Concerns Across Multiple Reviewers**

### **CLARIFICATION-1: Fairness when comparing with self-supervised baselines**
CWA-MSN is indeed a weakly supervised learning method because it leverages perturbation labels.

We intended to include methods from different groups to ensure a comprehensive and fair evaluation.

Weakly Supervised Group: CellCLIP, ViT-WSL, MolPhenix, and CLOOME use proxy-label supervision, making them appropriate references as they also leverage perturbation-level supervision.

Self-Supervised Group: OpenPhenom, Phenom-1, and Phenom-2 are state-of-the-art models on RxRx3-core, positioning our work within the broader field. OpenPhenom serves as the best publicly available SSL baseline.

Crucially, CWA-MSN demonstrates its competitive advantage by achieving strong performance with superior data and parameter efficiency, even against the most relevant weakly supervised baselines (e.g., CORUM: .386 for CWA-MSN vs. .354 for CellCLIP).

### **CLARIFICATION-2: Detailed comparison with SSLProfiler [1]**
We detail the difference between our design and [1] in the following:

1. Problem aimed

[1] targets within-well variation by aligning multiple site images of the same well. Our method targets cross-well and cross-plate variation, the dominant source of batch effects in large-scale Cell Painting datasets.

2. Architectural

[1] leverages DINOv2 with an auxiliary local-aggregation branch to encourage consistency across sites within a well, while our method adopts a Masked Siamese Network with two branches that align representations to a fixed set of prototypes, avoiding additional architectural components and multiple auxiliary losses.

In addition, our method explicitly selects wells with the same perturbation across plates and positions and aligns them directly in prototype space, removing the need for post-hoc merging procedures such as site interpolation or position-specific cross-plate merging.

Finally, [1] assumes identical perturbations across corresponding well positions, which conflicts with evidence that well position itself is a major source of batch effects [2].

Our method provides a more robust mechanism for mitigating batch effects and capturing biologically meaningful perturbation signals.

3. Supervision signal

While [1] leverages multiple losses (DINOv2 self-distillation, masked patch prediction, site-contrastive auxiliary loss), these competing objectives require careful gradient balancing.

Especially, site-level contrastive terms can pull representations toward site-specific features, creating instability.

In contrast, we use a single, unified objective based on prototype-distribution alignment: masked anchor views are trained to match the prototype structure of unmasked targets, with entropy regularization ensuring stability.

**Summary**

While [1] reduces within-well site variation using multiple DINOv2 losses as local aggregation, CWA-MSN directly targets cross-well consistency with an efficient, prototype-based masked-siamese objective, providing representations of perturbation-induced phenotypes while avoiding technical batch effects.

[1] Dai et al., *Self-supervised Representation Learning with Local Aggregation for Image-based Profiling - CVPR CVDD* (2025)

[2] Moshkov et al., *Learning Representations for Image-based Profiling of Perturbations - Nat. Commun.* (2024)

### **CLARIFICATION-3: Novelty concern**
We emphasize our novelty as following:

CWA-MSN is not intended as a direct application of MSN to Cell Painting data; rather, it incorporates novel sampling strategy and design choices well-suited for morphological profiling.

1. We select MSN over alternatives such as DINOv2 because its prototype-aligned masked-Siamese objective offers a stable learning signal with minimal augmentation requirements, which is essential in microscopy where strong natural-image augmentations such as Gaussian Blur can distort subtle perturbation-dependent features.

2. Incorporating MSN’s masking mechanism enables efficient training by removing the need for self-reconstruction (MAE) or contrastive sampling, which significantly reduces computational overhead and fits the constraints of modeling perturbation effects under limited supervision and realistic compute budgets.

3. Unlike prior work such as Phenom-1/2, which relies on ultra large datasets and models, our lightweight architecture (22M parameters, 0.2M images) along with critical sampling strategy intentionally ease the learning of perturbation effects.

4. Most importantly, CWA-MSN introduces the cross-well alignment strategy, a domain-specific contribution that forces representations from wells sharing the same perturbation to align across plates, directly mitigating batch and well-specific variation while preserving true morphological signal.

These components collectively make CWA-MSN a purpose-built, efficient, and biologically grounded method instead of a simple reuse of MSN.

---

### Author Response · Authors · 2025-12-02
**Author Clarification Regarding the Identity-Leakage Event and Assurance of Non-Collusion**

We write to provide a final summary for Paper 10496 to ensure that the decision accurately reflects the scientific consensus established prior to the recent double-blind system issues.

Although the system interruption obscured the updated scores, our rebuttal was demonstrably effective: we directly addressed the reviewers’ concerns, leading Reviewer WnET to raise their score from **4 to 6**.

**Precision of the Rebuttal Discussion**

As documented in the discussion, the exchange leading to the score increase was careful and focused purely on technical matters, with no connection to author identity. Reviewer WnET examined in depth the distinction between performance gains and batch-effect removal, as well as the architectural differences relative to SSLProfiler.

**Recognition of the Contribution**

Reviewers recognized our contribution to cell-level feature learning, noting the value of our cross-well sampling, data-efficient performance, and consistent gene and compound retrieval improvements. We believe they would likely have responded positively to our detailed rebuttal had the review period not been halted.

**Integrity of the Process**

We reiterate that we fully adhered to anonymity and all review policies; we **did not** attempt to exploit the system or access any prohibited information. Based on the focused, detailed, and technically grounded reviews, we believe that the reviewers likewise did not rely on any identity-related information in their assessments.

**Conclusion**

Given the reviewers’ recognition of the paper’s novelty and improvement, we respectfully request that the increased rating be honored and that the paper be considered for acceptance based on its demonstrated contribution to the biology field.

Sincerely,

The Authors

---

### Meta-Review · Area_Chair_h66S · 2026-01-07

**Summary:**

This paper proposes Cross-Well Aligned Masked Siamese Network (CWA-MSN), which aligns embeddings of cells under the same perturbation across different wells and integrates this into a masked-siamese framework to learn Cell Painting representations that are robust and efficient. The paper reports sizable gains on a gene-gene relationship retrieval benchmark (+29% vs OpenPhenom and +9% vs CellCLIP) while using much less data (0.2M vs 2.2M images) and a much smaller model (22M vs 1.48B parameters), and provides an anonymous code link.

**Reviewer Concerns:**

### Key reviewer concerns
- Novelty / positioning vs prior SSL: multiple reviewers felt the contribution is largely a domain-specific adaptation of masked-siamese/MSN-style methods, with unclear separation from closely related work (e.g., SSLProfiler).

- “Batch effect” claim vs supervision: concerns that using perturbation labels for positive pairing makes the method not purely self-supervised, and that the narrative (“remove batch effects”) did not cleanly match the objective.

- Need direct evidence that batch effects are reduced: requests to quantify batch effects (not only downstream retrieval).

- Scope, baselines, and sensitivity: requests for broader evaluation/generalization beyond RxRx3-Core, supervised baselines (e.g., SupCon), and sensitivity analyses (pair selection, ablations), plus reporting compute (FLOPs).


### and status after rebuttal/discussion
The rebuttal/discussion added concrete supporting evidence on two of the most important points: (i) batch-effect probing via plate prediction (macro-F1 for plate identity is slightly lower for CWA-MSN than alternatives, suggesting less plate information in embeddings) and (ii) compute reporting via FLOPs (CWA-MSN is far cheaper per image than MAE while remaining competitive) . Reviewer WnET explicitly stated their primary concerns were thoroughly addressed.

**Reviewer Scores:**

Reviewer WnET: post-discussion sentiment supports 6 (Weak accept)  (authors report 4 -> 6).

Reviewer aUNZ: with the added batch-effect probe + clearer framing, I expect 4 -> 5 (borderline).

Reviewer MPNE: novelty concerns likely remain, but added quantification/clarifications could move to 4 ->5, or unchanged

Reviewer sW6k: FLOPs + baseline clarifications address part of the feedback, likely 4 -> 5/6, or unchanged

---

### Decision · Program_Chairs · 2026-01-26

Reject